# Differentially Aquaporin 5 Expression in Submandibular Glands and Cerebral Cortex in Alzheimer’s Disease

**DOI:** 10.3390/biomedicines10071645

**Published:** 2022-07-08

**Authors:** Desiree Antequera, Laura Carrero, Victoria Cunha Alves, Isidro Ferrer, Jesús Hernández-Gallego, Cristina Municio, Eva Carro

**Affiliations:** 1Group of Neurodegenerative Diseases, Hospital Universitario 12 de Octubre Research Institute (imas12), 28041 Madrid, Spain; eeara@yahoo.es (D.A.); carrero.olivas@gmail.com (L.C.); vdacunhaalves@gmail.com (V.C.A.); jhgallego@salud.madrid.org (J.H.-G.); 2Network Center for Biomedical Research in Neurodegenerative Diseases (CIBERNED), ISCIII, 28031 Madrid, Spain; 8082ifa@gmail.com; 3Bellvitge Biomedical Research Institute (IDIBELL), 08908 Hospitalet de Llobregat, Spain; 4Department of Pathology and Experimental Therapeutics, University of Barcelona, 08907 Hospitalet de Llobregat, Spain; 5Institute of Neurosciences, University of Barcelona, 08035 Barcelona, Spain; 6Department of Neurology, Hospital Universitario 12 de Octubre, 28041 Madrid, Spain; 7Department of Medicine, Faculty of Medicine, Complutense University of Madrid, 28040 Madrid, Spain; 8Neurobiology of Alzheimer’s Disease Unit, Chronic Disease Programme, Instituto de Salud Carlos III, 28222 Majadahonda, Spain

**Keywords:** Alzheimer’s disease, aquaporins, submandibular gland, astrocytes, neurons

## Abstract

Impaired brain clearance mechanisms may result in the accumulation of aberrant proteins that define Alzheimer’s disease (AD). The water channel protein astrocytic aquaporin 4 (AQP4) is essential for brain amyloid-β clearance, but it is known to be abnormally expressed in AD brains. The expression of AQPs is differentially regulated during diverse brain injuries, but, whereas AQP4 expression and function have been studied in AD, less is known about AQP5. AQP5 functions include not only water transport but also cell migration mediated by cytoskeleton regulation. Moreover, AQP5 has been reported to be expressed in astrocytes, which are regulated after ischemic and traumatic injury. Additionally, AQP5 is particularly abundant in the salivary glands suggesting that it may be a crucial factor in gland dysfunction associated with AD. Herein, we aim to determine whether AQP5 expression in submandibular glands and the brain was altered in AD. First, we demonstrated impaired AQP5 expression in submandibular glands in APP/PS1 mice and AD patients. Subsequently, we observed that AQP5 expression was upregulated in APP/PS1 cerebral cortex and confirmed its expression both in astrocytes and neurons. Our findings propose AQP5 as a significant role player in AD pathology, in addition to AQP4, representing a potential target for the treatment of AD.

## 1. Introduction

Alzheimer’s disease (AD) is a progressive incurable neurodegenerative disease of the brain and has become the most common form of dementia in the elderly [1,2]. The neuropathological hallmarks of AD include extracellular plaques of amyloid-beta (Aβ), intraneuronal neurofibrillary tangles of hyperphosphorylated tau protein, neuroinflammation synaptic destruction, neuronal death, and brain atrophy [3,4,5,6]. However, accumulating evidence suggests that AD may manifest with pathology and/or dysfunction in other tissues, not restricted to the brain.

A number of studies indicates that patients with AD have systemic manifestations accompanying nervous system dysfunction, which suggests that the disease affects both the brain and the peripheral organs. Several studies have reported changes in the function and morphology of salivary glands in AD [7,8,9,10]. Saliva is a biofluid produced mainly by three pairs of major salivary glands, the submandibular, parotid, and sublingual glands [11]. Saliva is crucial for the maintenance of the oral health and function, as this secretion has antibacterial properties based on its antimicrobial protein content [12]. Salivary secretion is controlled by the parasympathetic nervous system through acetylcholine action on the acinar muscarinic receptors in salivary glands [13,14,15]. This modulation leads to intracellular calcium concentration and causes the secretion of water, mainly through a water channel called aquaporin (AQP). AQPs are a family of transmembrane channel proteins that facilitate transport of water across cellular membranes. AQPs have specific tissue and cellular expression and play critical roles in important physiological functions, including urine concentration, lactation, and formation of tears, sweat, and saliva [16,17]. To date, at least 13 AQP members have been found in mammals [18], and at least AQP1, AQP3, AQP4, AQP5, and AQP8 are known to be present in the salivary glands of humans and rodents, participating in saliva secretion [18]. AQP5 is mostly abundant in the salivary glands where it is located on the apical side of the acinar cells, in the intercellular secretory capillary, and in the luminal cytoplasm of the interstitial conduit [18]. Decreased AQP5 expression was reported in several illness, including Sjögren’s syndrome [19], and after radiation [20]. However, no information is available regarding the relationship between salivary AQP5 and AD.

AQPs also are expressed in the brain [21,22,23]. Meanwhile, AQP1, 4, and 9 serve multiple functions, including regulation of water movement, astrocyte function, production of cerebrospinal fluid, or even controlling apoptosis pathways in brain-related disorders [21,23,24,25,26,27], the functions of AQP5 in the brain are still unclear. AQP5 has been detected in primary cultures of astrocytes and neurons and also in rat brain in normal, ischemia, and traumatic injured conditions [28]. Due to its localization on both the membrane and in the cytoplasm of astrocytes, these authors suggested that AQP5 may be involved in multiple functions in astrocytes [28]. In this study, authors demonstrated that AQP5 expression was downregulated under ischemia injury but was upregulated after scratch-wound injury [28]. These results indicated that AQP5 could possibly be involved in astrocyte activation and glial scar formation after traumatic brain injury. These findings are in accordance with previous data on AQP5 role in cell proliferation and cell migration in cancer correlating with pathological manifestations and poor prognosis [29,30,31,32].

In this study, we aimed to evaluate the expression of AQP5 in submandibular glands and the brain during AD progression. We analyzed submandibular gland and cerebral cortex samples of a mouse model of cerebral amyloidosis (APP/PS1 transgenic mice) and AD patients. We found a differential AQP5 expression in submandibular glands depended on the pathology progression being reduced in 6-month-old APP/PS1 mice and increased at 12 months of age, whereas in cerebral cortex AQP5 expression is increased at both ages. In AD patients, AQP5 expression in submandibular glands showed a similar reduction than that observed in 6-month-old APP/PS1 mice, whereas at cortical level only AQP5 mRNA expression increased.

## 2. Material and Methods

### 2.1. Mice Samples

In this study, we used 6- and 12-month-old male double transgenic APP/PS1 mice, a cross between Tg2576 (overexpressing human APP695) and mutant PS1 (M146L), and their wild type (wt) littermates (nontransgenic) as control group. They were housed at our inbred colony (Hospital Universitario 12 de Octubre Research Institute). Animals were sacrificed by being deeply anaesthetized and perfused transcardially either with saline for biochemical analysis or with 4% paraformaldehyde (PFA) in 0.1 M phosphate buffer (PB), pH 7.4 for immunohistochemical analysis. All animals received care in compliance with the Council Directive 2010/63/UE of 22 September 2010 and ARRIVE guidelines (2020), and procedures were approved by the Hospital 12 de Octubre Ethic Committee. The number of animals used in each experiment is indicated in the legend of the corresponding figure.

### 2.2. Human Samples

We included post-mortem tissues from donors with a diagnosis of AD and healthy control individuals. Subjects were selected based on post-mortem diagnosis of AD according to neurofibrillary tangle pathology and Aβ plaques [33]. Absence of neuropathological change was reported in control participants. Subjects’ consent was obtained following the Declaration of Helsinki and the approval of the research ethics committee of abovementioned responsible institutions. The Institute of Neuropathology Brain Bank IDIBELL-Hospital Universitari de Bellvitge (Hospitalet de Llobregat, Spain), the University of Alabama at Birmingham-CHTN (Birmingham, AL, USA) and Banner Sun Health Research Institute, Arizona (Surprise, AZ, USA) supplied frozen submandibular gland samples. We evaluated 26 samples divided into two groups, as presented in Table 1.

Frozen orbitofrontal cortex samples were supplied by the Institute of Neuropathology Brain Bank IDIBELL-Hospital Universitari de Bellvitge (Hospitalet de Llobregat, Spain). A total of 73 samples were used in this study, as presented in Table 2.

### 2.3. Western Blot

Proteins were isolated from mouse and human submandibular gland and cerebral cortex samples by standard methods. Briefly, submandibular glands and cerebral tissues were homogenized in lysis buffer NP-40 (50 mM Tris-base pH 7.4, 150 mM NaCl, 0.5% Nonidet P-40, 1 mM EDTA) and protease and phosphatase inhibitors (ROCHE cOmplete™ Protease Inhibitor Cocktail-Roche, Basel, Switzerland) were centrifuged for 15 min at 10,000 rpm at 4 °C. Protein estimation from lysates was determined using the BCA assay (Pierce BCA Protein Assay Kit, Thermo Fisher Scientific, Waltham, MA, USA). Proteins were separated in a precast 4-20% Tris-HCl (CriterionTM TGX Stain-FreeTM Precast Gels, BioRad Laboratories, Hercules, CA, USA) and transferred to polyvinylidene fluoride (PVDF) membranes (BioRad Laboratories). Primary antibodies utilized were rabbit monoclonal to human Aquaporin 5 (ab92320, Abcam, Cambridge, UK) and rabbit polyclonal to mouse Aquaporin 5 (ab78486, Abcam, Cambridge, UK). Membranes then were incubated for 1 h with HPR mouse monoclonal antibody against β-actin (ab49900, Abcam, Cambridge, UK) and used as protein loading, and immunocomplexes were revealed by an enhanced chemiluminescence reagent (ECL Clarity; BioRad Laboratories). Densitometric quantification was performed with Image Studio Lite 5.0 software (Li-COR Biosciences, Lincoln, NE, USA). Protein bands were normalized with β-actin levels, as a control loading protein, and the measurements were expressed as a percentage of the control group.

### 2.4. Immunohistochemistry

Submandibular glands tissue was fixed for 24 h in 4% PFA by immersion. Then, submandibular gland samples were paraffin-embedded, and free-floating 4 µm thick sections were obtained using a cryotome (Leica, Wetzlar, Germany).

Cerebral cortex tissue was fixed for 24 h in 4% PFA by immersion. Then, brain samples were OCT embedded at −80 °C, and free-floating 30 µm thick sections were obtained using a cryostat (Leica, Wetzlar, Germany).

Samples were pre-incubated with 0.2M citrate for 20 min and heated in the microwave at 100 °C for 15 min. All primary antibodies were diluted in PB 0.1 M containing 0.5% bovine serum albumin and 0.5% Triton X-100. The following primary antibodies were used: Rabbit monoclonal to human AQP5 (ab92320, Abcam, Cambridge, UK) and rabbit polyclonal to mouse AQP5 (ab78486, Abcam, Cambridge, UK). After overnight incubation, primary antibody staining was revealed using the avidin-biotin complex method (PK6101 Rabbit IgG, VECTASTAIN Elite ABC Kit, Vector Laboratories, Burlingame, CA, USA). The sections were counterstained with Vector hematoxylin (H3401, Vector Laboratories, Burlingame, CA, USA) and 3,3′-diaminobenzidine (DAB) chromogenic reaction (SK-4100, Vector Laboratories, Inc). Finally, the slices were mounted with DPX (Panreac Quimica, Barcelona, Spain). Images were captured using a light microscope Zeiss Axiocam ERc5s camera and Zen 2012 software on a Zeiss Axiocam Scope (Carl Zeiss Microimaging, GmbH, Oberkochen, Germany). Immunohistochemistry analysis was performed measuring intensity of AQP5 signal from at least 7 different animals per group in cortex and 10 animals per group in submandibular glands. We analyzed 5 slides in each animal. Data were presented as the percentage of AQP5^+^ area stained with AQP5 using the Image J (v 1.53p, NIH, Bethesda, MD, USA). The exact number of animals used in each experiment is indicated in the legend of the corresponding figure.

For immunofluorescence, one series of sections was used for double-labelling experiments. We used rabbit monoclonal to human AQP5 (ab92320, Abcam), rabbit polyclonal to mouse AQP5 (ab78486, Abcam), mouse monoclonal to GFAP (G3893, Sigma-Aldrich, Merck Life Science, Madrid, Spain), and mouse monoclonal to NeuN (ab104224, Abcam, Cambridge, UK) as primary antibody, and they were revealed using fluorescence-conjugated secondary antibodies from Life Technologies: Alexa Donkey anti-mouse 488 (A21202) and Alexa Goat anti-rabbit 555 (A27039). Finally, the slices were mounted with Immunoselect Antifanding Mounting Medium DAPI (SCR-038448, BioTrend, Köln, Germany). Fluorescent images were obtained by laser confocal microscopy (Zeiss-LSM 510, Oberkochen, Germany) using the excitation lasers 405, 488, and 568 for DAPI and green and red fluorescence, respectively. For the acquisition of fluorescence images, a fixed exposure was maintained for all samples in all the experiments. All images were analyzed using the Volocity 3D Image Analysis. At least 2 images from 4 different individuals per group were analyzed.

### 2.5. RNA Extraction and Quantification

Total RNA from mouse and human cerebral cortex was extracted using NZYol Reagent (NZYTech, Lisboa, Portugal) according to the manufacturer´s protocol. RNA concentration was measured in a Nanodrop One Spectrophotometer (Thermo Fisher Scientific, Waltham, MA, USA). A total of 1 ug was used for fist-strand complementary DNA synthesis with iScript cDNA Synthesis kit (Bio-Rad Laboratories, Hercules, CA, USA). Quantitative real-time PCR (qRT-PCR) was performed on a Roche LightCycler 480 II (Roche Diagnostics, Basel, Switzerland) instrument using NZYSupreme qPCR Green Master mix (NZYTech, Lisboa, Portugal) according to the manufacturer´s recommendations. Primer sequences are listed in Table 3. HPRT mouse and human were used as endogenous reference. For relative quantification the Ct value of each gene was normalized with its endogenous reference gene using the 2^−ΔCt^ formula. Then, gene expression change was compared to the control sample expression using the 2^−ΔCt^ formula (2^−ΔΔCt^ formula, Ct = threshold cycle).

### 2.6. Data and Statistical Analysis

Data analysis was conducted using GraphPad Prism 6.01 (GraphPad Software, San Diego, CA, USA) software. All data are expressed as mean ± standard error of the mean (SEM). Statistical comparisons between two groups were performed with Student’s *t*-test or Mann–Whitney test as appropriate. Multiple comparisons were calculated by one-way ANOVA followed by Dunnet correction. In all cases, statistical significance was set at *p* < 0.05.

## 3. Results

### 3.1. Expression of AQP5 in Submandibular Glands from APP/PS1 Mice and AD Patients

AQP5 is particularly abundant in the salivary glands; therefore, we first performed an immunohistochemistry analysis on submandibular glands from 6- and 12-month-old APP/PS1 and wt mice (Figure 1A,B). We found that AQP5 immunoreactivity, localized in epithelial cells of serous acini, was significantly reduced in submandibular glands from 6-month-old APP/PS1 mice compared with age-matched wt mice (Figure 1A). To confirm these results, we also assessed AQP5 levels by western blot of submandibular glands from these mice groups, and we observed a similar decrease in AQP5 levels at 6-month-old in APP/PS1 mice compared with wt mice (Figure 1C). However, when we analyzed 12-month-old mice groups, we detected a contrary tendency. AQP5 immunostaining was significantly higher in submandibular glands from 12-month-old APP/PS1 mice compared with age-matched wt mice (Figure 1B). Moreover, western blot analysis confirmed that APP/PS1 mice showed higher AQP5 levels in their submandibular glands compared with wt mice at 12-month-old age (Figure 1D).

Then, we evaluated AQP5 expression in human submandibular glands by immunohistochemistry, and we found AQP5 localization in epithelial cells of serous acini in salivary gland similar to what we previously detected in mouse samples (Figure 2A). We observed a significant reduction in AQP5 expression in submandibular glands from AD patients compared with healthy control subjects, which was a diminishing comparable to that observed 6-month-old APP/PS1 mice (Figure 2A). This decrease also was found when AQP5 levels were determined by western blot (Figure 2B). Our results showed a drastic reduction in AQP5 expression in AD submandibular glands compared with that observed in control individuals (Figure 2B).

### 3.2. Expression of AQP5 in Cerebral Cortex from APP/PS1 Mice and AD Patients

As AQP5 expression was previously reported in mouse and rat cerebral cortex, we first confirmed mRNA expression of AQP5 in brain tissue in both experimental mouse models by RT-PCR. As we expected, low expression of AQP5 mRNA was found in cerebral cortex samples (Figure 3A). However, the levels of AQP5 mRNA increase with age in APP/PS1 mice and were significantly higher in 12-month-old APP/PS1 mice compared with those detected in age-matched wt mice (Figure 3A). The expression of AQP1, 3, 4, 8, and 9 in these mouse brain samples also was examined by RT-PCR (Figure 3A). While the expression of AQP1, 3, 8, and 9 expression was detected but unchanged in these samples, the expression of AQP4 mRNA significantly increased in 12-month-old APP/PS1 mice compared with age-matched wt mice, similarly as we had seen regarding AQP5 (Figure 3A).

Next, we explored levels of AQP5 protein in cerebral cortex from 6- and 12-month-old mice groups. Western blot analysis confirmed higher AQP5 levels in cortical lysate samples from transgenic mice compared with wt mice in both 6- (Figure 3B) and 12- (Figure 3C) month-old groups.

Then, we examined the expression of AQP5 in cerebral cortex from these mice groups by immunostaining. Immunohistochemical assay revealed a significant increase in AQP5 staining intensity in cerebral cortex slides from 6-month-old APP/PS1 mice compared with age-matched wt mice (Figure 4A). In addition, cortical AQP5 signal intensity also increased significantly in 12-month-old APP/PS1 mice compared with age-matched wt mice (Figure 4B). Together, these results from mRNA expression, western blot, and immunostaining assays confirmed that AQP5 expression was upregulated in APP/PS1 mice brain compared with age-matched wt mice.

When we explore cell populations with AQP5-positive signal, we found that AQP5 immunoreactivity was present in GFAP-positive and NeuN-positive cells in mouse cerebral cortical slices by double immunofluorescence, corresponding to astrocytes and neurons, respectively. Our findings are in agreement with previously published data reporting AQP5 expression in GFAP-positive astrocytes in rat cerebral cortical tissue [28]. Our results revealed scarce reactive astrocytes expressing GFAP in 6-month-old wt mice; however activated astrocytes were visualized in cortical sections of 6-month-old APP/PS1 mice (Figure 4C). Moreover, significant AQP5 colocalization was observed in GFAP-positive astrocytes in transgenic mice being barely detectable in wt mice (Figure 4C). As we expected, astroglial activation enhanced in AD advance stages can be observed in 12-month-old APP/PS1 mice by the presence of abundant GFAP-positive astrocytes in their cerebral cortex compared to aged-matched control mice (Figure 4D). In the cortical tissue of these transgenic mice, we perceived similar AQP5 colocalization in GFAP-positive astrocytes than that observed in 6-month-old APP/PS1 mice.

Double immunofluorescence experiments showed that AQP5 and NeuN colocalized in the cerebral cortex of APP/PS1 and wt mice, as were shown in 6- and 12-month-old mice (Figure 4E and 4F, respectively). In these NeuN-positive cells, AQP5 immunoreactivity was seen as a dispersed punctate precipitate contained within the cytoplasm in both APP/PS1 and wt mice without appreciating significant differences throughout the aging of the mice.

Additionally, we investigated the expression of AQP5 in cerebral cortical samples from AD patients. As we performed above with the mice studies, we first measured mRNA expression of AQP5 in cortical tissue from brain donors with AD pathology and healthy subjects by RT-PCR. Our findings revealed very low expression of AQP5 mRNA in human cortical samples; however, we found that mRNA AQP5 expression significantly increased not only in Braak stages III/IV and V/VI compared with control samples but also with Braak stages I/II (Figure 5A). The expression of AQP1, 3, 4, 8, and 9 also was determined (Figure 5A). Consistently, the expression of all these AQPs was significantly enhanced in cerebral cortical tissues from brain donors with last AD stages compared to control samples (Figure 5A).

When we explored AQP5 protein expression in these brain samples, it was practically undetectable using western blot (data not shown). Alternatively, we explored AQP5 expression in human brain with immunohistochemistry. As shown in Figure 5B and as we expected, weak AQP5 immunoreactivity was found in cortical samples from both control subjects and AD patients, and quantification of the AQP5 signal revealed similar expression between control and AD samples (Figure 5B).

## 4. Discussion

In the present study, we investigated the differences in AQP5 distribution and expression in submandibular glands and cerebral cortex during AD progression using APP/PS1 mice and AD patient samples. We found that AQP5 was differently expressed in submandibular glands and depended on the AD pathology progress being reduced in 6-month-old APP/PS1 mice and increased at 12 months of age. However, AQP5 expression increased in cerebral cortex in both 6- and 12-month-old mice compared to aged-matched control mice. In AD patients, AQP5 expression in submandibular glands showed a similar reduction than that observed in 6-month-old APP/PS1 mice, whereas at cortical level only the expression of AQP5 mRNA was enhanced.

We observed robust AQP5 expression in submandibular glands accordingly with previous studies that showed its expression in the acinar cells involved in the normal physiology maintenance of the salivary gland in rodents and humans [34,35]. Previous studies reported that expression of AQP5 decreased in the salivary glands under pathological situations, including in response to radiation [20], in Sjögren syndrome patients [19], in diabetes-induced rats [36], or associated with systemic inflammation after lipopolysaccharide (LPS) stimulation in mice [37]. Conversely, an early study described increased AQP5 protein expression in submandibular glands from NOD mice, a well-described mouse model for the study of the autoimmune exocrinopathy that is prevalent in patients with Sjögren’s syndrome, even with a significant decrease in the salivary flow rate [38].

However, less is known about the relationship between salivary AQP5 expression and neurodegeneration, and particularly, no studies have been reported associated with AD.

Numerous scientific publications indicate that AD patients have systemic manifestations accompanying nervous system dysfunction, which suggests that the disease affects both the brain and peripheral organs, including the salivary glands [8,9,10,39]. Salivary gland dysfunction in AD patients was associated with salivary redox imbalance linked with chronic inflammation [10]. Recently, it has been described as a hypothetical mechanism for LPS-induced downregulation of AQP5 [40]. In our present study, we showed that in submandibular glands from 6-month-old APP/PS1 mice AQP5 protein levels are significantly decreased compared to aged-matched control mice. An increasing body of evidence links systemic inflammation to AD development [41]. Since plasma levels of interleukin (IL)-1β, IL-6, tumor necrosis factor α (TNF-α), and macrophage inflammatory protein (MIP)-2 significantly increased in 6-month-old APP/PS1 mice compared with wild-type control mice [42], we propose the hypothesis by which systemic inflammatory signaling may induce AQP5 downregulation in submandibular glands in these mouse models of AD. Similarly, elevated levels of inflammatory proteins also have been reported in the plasma of AD patients [43,44]. In our study, submandibular glands from AD patients also displayed significantly lower AQP5 levels, reinforcing the theory that peripheral inflammation may trigger detrimental effects on submandibular AQP5 levels and salivary gland function.

In addition, AQP5 protein levels in the submandibular gland is regulated by the parasympathetic nerves/M3 muscarinic receptor [45]. Downregulation of submandibular gland AQP5 by denervation of the parasympathetic nerve was reported [46,47]. In a recently published study, we described a significant reduction in M3 muscarinic receptor levels in submandibular glands from APP/PS1 mice and AD patients [8]. Based on all these findings, we proposed that AQP5 levels in submandibular glands may be regulated by peripheral and central mechanisms, inducing changes in AQP5 levels in the salivary gland that depends on normal and pathologic conditions, such as, in the present situation, AD.

In contrast, submandibular glands from 12-month-old APP/PS1 mice displayed an opposite pattern, exhibiting markedly higher AQP5 protein levels. Reversible post-translational modifications, such as phosphorylation and ubiquitination, play important roles in regulating the expression level of AQP5 [40]. Several studies suggest that AQP5 may be degraded by autophagy in submandibular glands [47,48]. Accumulating evidence has indicated that impaired autophagy is closely correlated with AD pathogenesis [49,50,51]. It has been proposed that the enhanced autophagy at the early stage of AD is protective in response to stress, but autophagic flux is ultimately compromised with AD progression [52]. Thus, based on these finding, we argue that lower AQP5 levels in submandibular glands from 6-month-old APP/PS1 mice also may be regulated by enhanced autophagy processes and that degradation diminished because of dysfunctional autophagy in these APP/PS1 mice at later AD stages. However, further studies will be required to confirm the mechanisms leading to the increased level of AQP5 protein in submandibular glands from 12-month-old APP/PS1 mice and to characterize the possible link existing between autophagy in AD submandibular glands and AQP5 levels.

In this study, we demonstrated that AQP5 expression was increased in cortical tissue from 6-month-old APP/PS1 mice, and this enhancement is higher in aged transgenic mice. However, we found discrepancies between protein and RNA abundance of AQP5 in cerebral cortex of postmortem samples of AD patients. Meanwhile, we detected a progressive increase in mRNA AQP5 expression in Braak stages III/IV and V/VI; levels of AQP5 protein were very scare in these cerebral cortical samples. We think that this effect may result in low mRNA AQP5 expression in brain tissue, and the correlation between expression levels of protein and mRNA in mammals is relatively low [53,54].

In brain samples, we confirmed different cellular localization of AQP5. Herein, we showed by co-immunostaining assays that AQP5 was localized in the cytoplasm of astrocytes and neurons. Our findings are consistent with prior studies performed in primary cell cultures and rat brain tissue showing that AQP5 was expressed in astrocytes and detected AQP5 mRNA in primary cultures of astrocytes and neurons [22,28]. AQPs have been studied in various brain pathological conditions, AQP4 being the most frequent. AQP4 is known to be involved in various astrocytic functions related to neurological disorders [55,56]. AQP4 expression was observed surrounding senile plaques in AD [57]. Additionally, perivascular AQP4 localization was associated with increased Aβ burden and increased Braak stage [55]. Glymphatic function is shown to reduce in mid- to late-stage AD due to the loss in polarity of AQP4 at the astrocyte end feet [55]. AQP4 expression is normally polarized insofar as it is expressed within the astrocytic end feet rather than in the astrocytic somata [58]. Increasing evidence indicates that astrocytes also express AQP1 under pathological conditions. AQP1 expression is augmented in cortical astrocytes at the early stage of AD [59], and AQP1-expressing fibrillary astrocytes might have a close association with Aβ deposition in the brains of advanced stages of AD [60]. Because we and others demonstrated AQP5 expression in astrocytes, we suggest that the reactive astrogliosis in AD could be responsible for the increased expression in AQP5 observed in APP/PS1 mice in our present study. However, the mechanism of regulation of the AQP5 expression in the brain, and particularly in astrocytes, has not been investigated. However, it was reported that the cAMP/PKA pathway, one of the major signal transduction pathways in astrocytes, is involved in downregulation of AQP5 [61], and this pathway is compromised in AD [62]. Therefore, impaired AMP/PKA pathway in AD also could be involved in the increase expression of AQP5 in astrocytes.

To our knowledge, this is the first study to describe AQP5 expression in submandibular glands and brain under AD pathology. In summary, herein we demonstrated that AQP5 expression in submandibular glands was reduced in 6-month-old APP/PS1 mice and AD patients but was upregulated in aged transgenic mice. We also observed increased AQP5 expression in cerebral cortex from APP/PS1 mice, mainly being localized in astrocytes and neuronal cells. Since AD is associated with astrocyte activation, we suggest that higher AQP5 expression found in APP/PS1 mice compared to wt mice could be associated with AD astrogliosis, indicating a potential role of AQP5 in astrocyte response to AD pathology. Our present findings suggest that changes in brain AQP5 expression are features of AD brain.

## Figures and Tables

**Figure 1 biomedicines-10-01645-f001:**
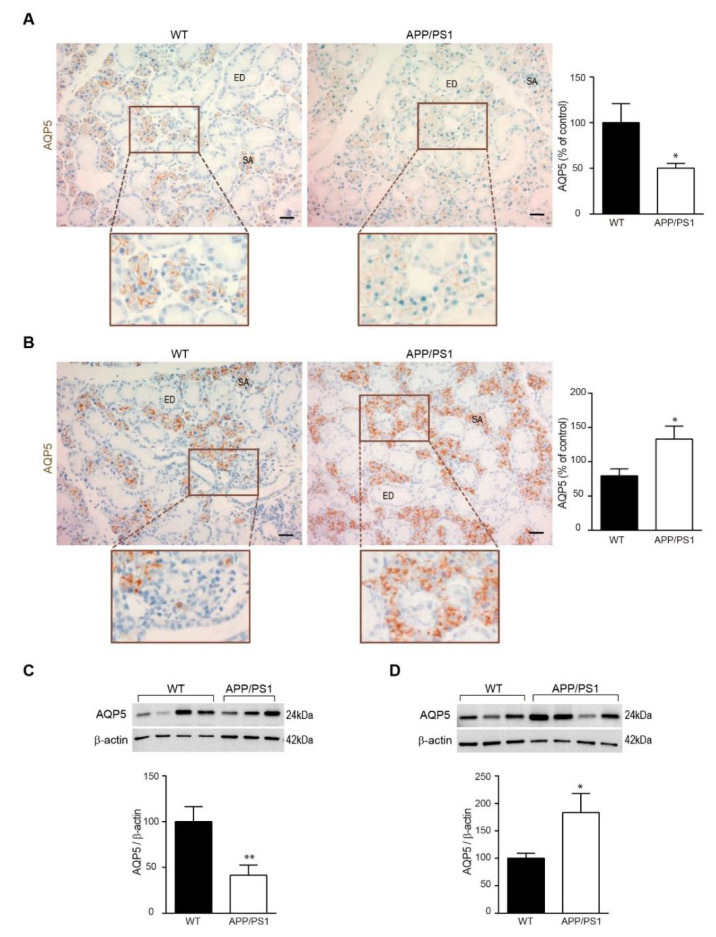
**Expression of AQP5 in mouse submandibular gland.** Localization of AQP5 in submandibular gland from APP/PS1 transgenic and nontransgenic mice was detected by immunostaining and western blot. (**A**,**B**) Representative AQP5 stained sections of submandibular glands from (**A**) 6- and (**B**) 12-month-old APP/PS1 and wt mice. All sections were counterstained with hematoxylin. In the lower histograms, quantification of the AQP5 signal (*n* = 6, per group) is shown. Scale bar = 20 μm. Inserts show higher magnification views. (**C**,**D**) Western blot analysis showing AQP5 levels in submandibular glands from (**C**) 6- and (**D**) 12-month-old APP/PS1 and wt mice (*n* = 12, per group). Representative western blots (upper panels) and histograms with their densitometric analysis (lower panels) are shown. Data are represented as mean ± SEM. Differences between groups were assessed using the Student’s *t*-test * *p* < 0.05; ** *p* < 0.01. SA: serous acini; ED: excretory duct.

**Figure 2 biomedicines-10-01645-f002:**
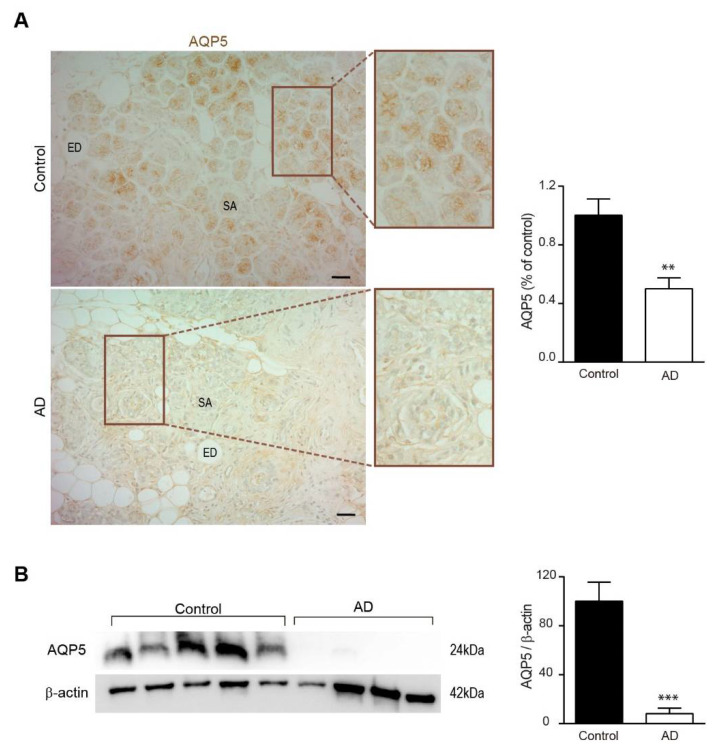
**Expression of AQP5 in human submandibular gland.** Localization of AQP5 in submandibular gland from AD and control post-mortem samples was detected by immunostaining and western blot. (**A**) Representative AQP5 stained sections of submandibular glands from AD and control post-mortem samples. All sections were counterstained with hematoxylin. In the right histogram, quantification of the AQP5 signal (Table 1) is shown. Scale bar = 20 μm. Inserts show higher magnification views. (**B**) Western blot analysis showing AQP5 levels in submandibular glands from AD and control post-mortem samples (*n* = 8, per group). Representative western blot and histogram with their densitometric analysis (right panel) are shown. Data are represented as mean ± SEM. Differences between groups were assessed using the Mann–Whitney test. ** *p* < 0.01; *** *p* < 0.001. SA: serous acini; ED: excretory duct.

**Figure 3 biomedicines-10-01645-f003:**
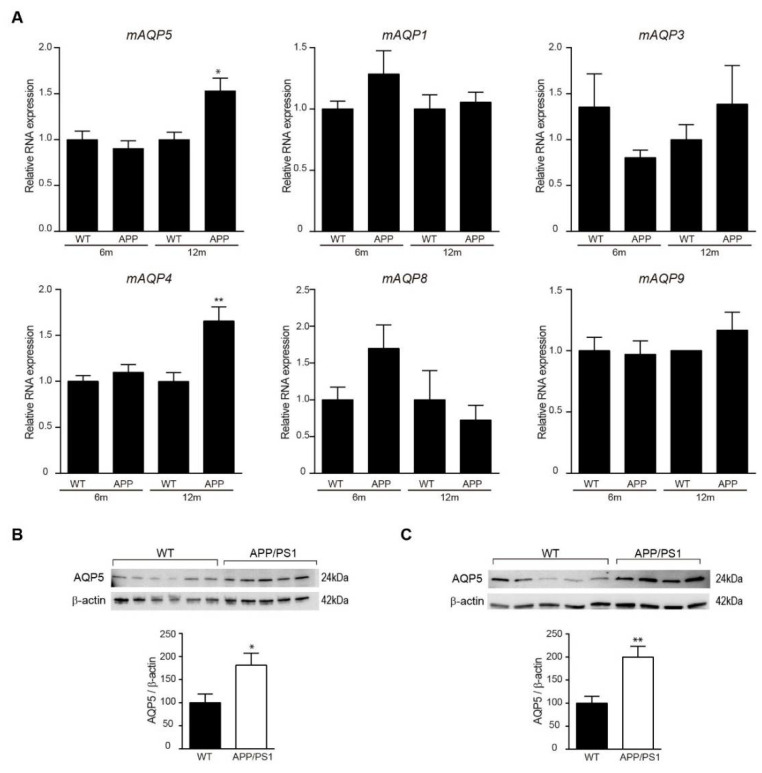
**Expression of AQP5 in mouse cerebral cortex.** Localization of AQP5 in cerebral cortex from APP/PS1 transgenic and nontransgenic mice was detected by RT-PCR and western blot. (**A**) mRNA expression of mouse AQP5, 1, 3, 4, 8, and 9 in brain samples from 6- and 12-month-old APP/PS1 and wt mice (*n*=12, per group). (**B**,**C**) Western blot analysis showing AQP5 levels in cerebral cortex from (**B**) 6- and (**C**) 12-month-old APP/PS1 and wt mice (*n* = 8, per group). Representative western blots (upper panels) and histograms with their densitometric analysis (lower panels) are shown. Data are represented as mean ± SEM. Differences between groups were assessed using the Student’s *t*-test. * *p* < 0.05; ** *p* < 0.01.

**Figure 4 biomedicines-10-01645-f004:**
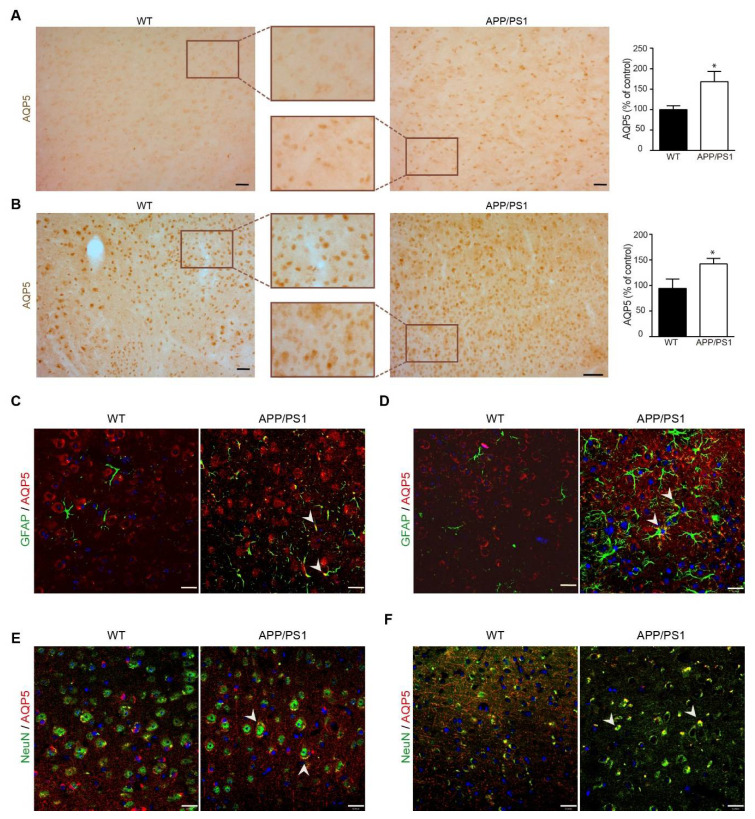
**Analysis of AQP5 expression in brain cell populations.** Localization of AQP5 in cerebral cortex from APP/PS1 transgenic and nontransgenic mice was detected by immunostaining and immunofluorescence. (**A**,**B**) AQP5 stained sections of cerebral cortex from (**A**) 6- and (**B**) 12-month-old APP/PS1 and wt mice. All sections were counterstained with hematoxylin. In the right histograms, quantification of the AQP5 signal (*n* = 8, per group) is shown. Inserts show higher magnification views. (**C**–**F**) Representative confocal images showing AQP5 (red) in astrocytes inmunostained with anti-GFAP (green) staining in (**C**) 6- and (**D**) 12-month-old APP/PS1 and wt mice. Representative confocal images showing AQP5 (red) in neurons inmunostained with anti-NeuN (green) staining in (**E**) 6- and (**F**) 12-months-old APP/PS1 and wt mice. White arrows indicate colocalization of the corresponding antibodies. Scale bar = 20 μm. Data are represented as mean ± SEM. Differences between groups were assessed using the Student’s *t*-test. * *p* < 0.05.

**Figure 5 biomedicines-10-01645-f005:**
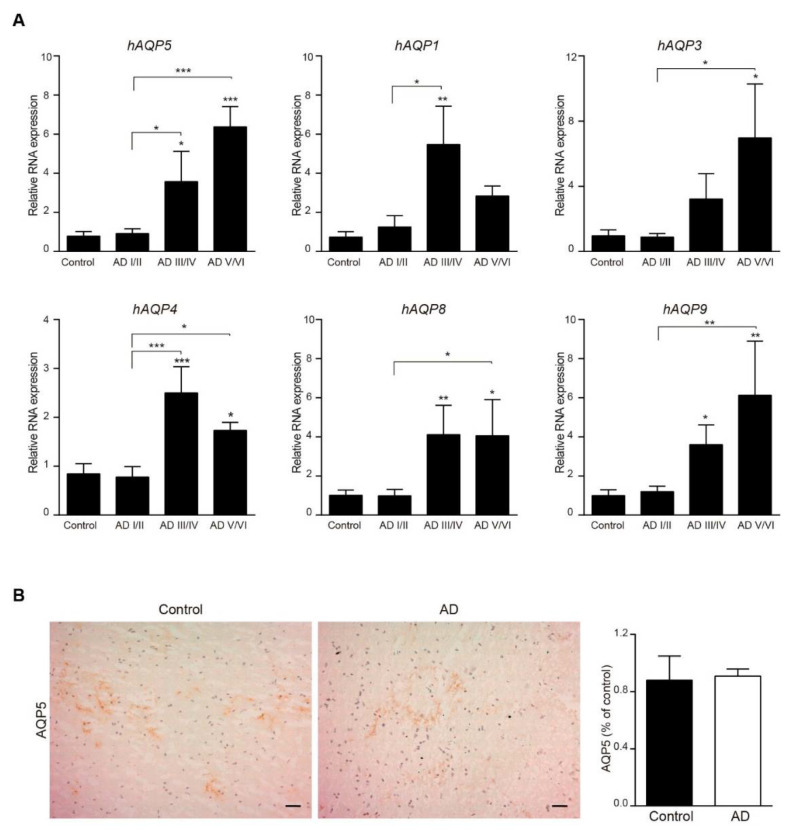
**Expression of AQP5 in human cerebral cortex.** Localization of AQP5 in cerebral cortex from AD and control post-mortem samples was detected by mRNA and immunostaining. (**A**) mRNA expression of human AQP5, 1, 3, 4, 8, and 9 in brain samples from AD and control patients (Table 2). (**B**) AQP5 stained sections of cerebral cortex from AD and control post-mortem samples. All sections were counterstained with hematoxylin. In the right histogram, quantification of the AQP5 signal (*n* = 8, per group) is shown. Scale bar = 20 μm. Data are represented as mean ± SEM. Differences between groups were assessed using the one-way ANOVA analysis followed by Dunnet correction. * *p* < 0.05, ** *p* < 0.01, *** *p* < 0.001.

**Table 1 biomedicines-10-01645-t001:** Demographic and clinical data of gland tissue donors.

	Control	AD
*n*	11	15
Sex (M/F)	6/5	9/6
Age Mean (SD)	62.27 (8.25)	79.07 (8.01)
Braak Stage	-	II–III: 5
		VI: 10

AD: Alzheimer’s disease; *n*: number; M: male; F: female; SD: standard deviation.

**Table 2 biomedicines-10-01645-t002:** Demographic and clinical data of brain tissue donors.

	Control	AD
*n*	26	47
Sex (M/F)	14/12	25/22
Age Mean (SD)	69.48 (15.84)	74.52 (9.10)
Braak Stage	-	I–II: 21
		III–IV:14 V–VI: 12

AD: Alzheimer´s disease; *n*: number; M: male; F: female; SD: standard deviation.

**Table 3 biomedicines-10-01645-t003:** Real-time PCR oligonucleotides.

Gene	Sequence (5´>3´)	
	Forward	Reverse
*mHPRT*	gttgggcttacctcactgct	taatcacgacgctgggactg
*mAQP1*	tcccctaactttcccctttg	agcacagggacaattccaag
*mAQP3*	cttgtgatgtttggctgtgg	aagccaagttgatggtgagg
*mAQP4*	ttccgttcgatcttcagagg	tatcagcccatttcccagag
*mAQP5*	Ttcaggaccatcccagaaag	taagatggcactcgacgaac
*mAQP8*	ttgctaccttggggaacatc	caatcagccctccaaatagc
*mAQP9*	tgcgacttttggtgtctctg	ttgaaccactccatccttcc
*hHPRT*	gaccagtcaacaggggacat	cctgaccaaggaaagcaaag
*hAQP1*	tttctgtttcctggcctcag	tccacaacttcaagggagtg
*hAQP3*	atgtgtgtgcatgtgtgtgc	tcccttgccctgaatatctg
*hAQP4*	gcatgtgattgacgttgacc	tgggtggaaggaaatctgag
*hAQP5*	tggctgccatcctttacttc	gctcatacgtgcctttgatg
*hAQP8*	aggttctggaatgcatctgg	ggccctttgtcttctcattg
*hAQP9*	tttatgtgggagcccagttc	gttttccaccagcaaaggac

HPRT: hypoxanthine-guanine phosphoribosyltransferase; AQP: aquaporin; m: mouse; h: human.

## Data Availability

The data that support the findings of this study are available from the corresponding author upon reasonable request.

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
