# Peer review of "Differentially Aquaporin 5 Expression in Submandibular Glands and Cerebral Cortex in Alzheimer’s Disease"

_biomedicines, 2022, doi:10.3390/biomedicines10071645_

Round 1
Reviewer 1 Report
In their manuscript, Antequera et al report the expression of AQP5 in submandibular glands and cerebral cortex from double transgenic APP/PS1 mice and from Alzheimer´s disease (AD) patients of different Braak stages. The study is entirely descriptive; yet it could be relevant to the field of Alzheimer´s as salivary gland dysfunction occurs in AD patients. However, the quantification and statistical analysis of the data obtained in the assays (Western blot, immunostaining and RT-qPCR) should be improved in order to increase confidence in the results.
1. Statistical analysis for multiple comparisons should be calculated using one-way ANOVA following by a Bonferroni, Tukey, Dunnett post-hoc analysis to correct for multiple comparisons.
2. Please, explain how image quantification was performed.
3. Relative quantification of the RT-qPCR data of AD versus controls should be best done by first normalizing the Ct value obtained for each target gene to the Ct value obtained for the reference gene using 2-∆Ct formula. Then, gene expression change in each AD Braak stage sample should be compared to the level of gene expression obtained in the control sample, again using 2-∆Ct formula; therefore 2-∆∆Ct. See, M W. Pfaffl (2001) Nucleic Acids Research and T D. Schmittgen & K J. Livak (2008) Nature Protocols.
4. The Western blot analysis shown in figure 1 C should be repeated in a few more animals as out of the four WT samples, two of them have relatively low AQP5 expression (comparable or lower to that of APP/PS1 mice) whereas the other two samples show higher expression.
Please, indicate in all the blots the molecular weight of the specific band.
Author Response
Please, see the attachment

Reviewer 2 Report
Comment 1:
On page (1), lines 31-32: the sentence need to be improved grammatically.
Comment 2:
On page (1), M&M section, mice samples: A diagram or a short description of the sex and number of mice used in each group should be added, (similar to that found in the Figure 3 description paragraph), in order to better understand the experimental protocol. When the major aims are to examine the pathogenesis of AD, using both sex of animals is crucial to make a conclusion. Thus, authors are encouraged to perform/include all experiments/data for both male and female to make a more meaningful study.
Comment 3:
On page (7), line 245: Reference studies should be avoided to be included in the results section, they can be included in the discussion section.
Reviewer 3 Report
The current manuscript is well designed and presented in a nice manner. However repreresentaion of microscopic images is very hard to follow. I would recommend to publish this manuscript after minor revision.
Round 2
Reviewer 1 Report
The manuscript is improved after revision.